# Combinatorial metabolic engineering using an orthogonal tri-functional CRISPR system

Jiazhang Lian [1,2], Mohammad HamediRad [1], Sumeng Hu[1] & Huimin Zhao [1,3]

Designing an optimal microbial cell factory often requires overexpression, knock-down, and knock-out of multiple gene targets. Unfortunately, such rewiring of cellular metabolism is often carried out sequentially and with low throughput. Here, we report a combinatorial metabolic engineering strategy based on an orthogonal tri-functional CRISPR system that combines transcriptional activation, transcriptional interference, and gene deletion (CRISPR-AID) in the yeast *Saccharomyces cerevisiae*. This strategy enables perturbation of the metabolic and regulatory networks in a modular, parallel, and high-throughput manner. We demonstrate the application of CRISPR-AID not only to increase the production of β-carotene by 3-fold in a single step, but also to achieve 2.5-fold improvement in the display of an endoglucanase on the yeast surface by optimizing multiple metabolic engineering targets in a combinatorial manner.

[1] Department of Chemical and Biomolecular Engineering, Carl R. Woese Institute for Genomic Biology, University of Illinois at Urbana-Champaign, Urbana, IL 61801, USA. [2] College of Chemical and Biological Engineering, Zhejiang University, Hangzhou 310027, China. [3] Departments of Chemistry, Biochemistry, and Bioengineering, University of Illinois at Urbana-Champaign, Urbana, IL 61801, USA. Correspondence and requests for materials should be addressed to H.Z. (email: zhao5@illinois.edu)

Microbial cell factories have been increasingly engineered to produce fuels, chemicals, and pharmaceuticals using various renewable feedstocks[1, 2]. However, microorganisms have evolved robust metabolic and regulatory networks to survive and grow in specific environments rather than to synthesize the products of industrial interest. Therefore, metabolic engineering of the producing microorganisms is required to rewire the cellular metabolism, i.e., to enhance the supply of the precursor metabolites[3–5], to maximize fermentation titer, yield, and productivity for commercially viable processes. To perturb the extensive regulation and complex interactions between metabolic pathways, researchers often need to modify multiple metabolic engineering targets with different modes of regulation, such as to increase expression of genes encoding rate-limiting enzymes, decrease expression of essential genes, and remove expression of competing pathways[1]. Researchers should be able to control a full spectrum of expression profiles for multiple genes of interest simultaneously. Unfortunately, such rewiring of cellular metabolism is often carried out sequentially and with low throughput, which is largely due to the lack of facile and multiplex genome engineering tools. Homologous recombination based gene replacement is commonly used for genome engineering of the producing microorganisms, but suffers from low efficiency and throughput and is labor and time intensive[6]. Consequently, genome engineering targets are mainly tested individually or in a few combinations. However, due to our limited knowledge on the regulation of cellular metabolism, it is highly desirable to test more metabolic engineering targets in combinations, particularly for those with synergistic interactions. Therefore, development of a combinatorial metabolic engineering strategy to modify the host genome in a modular, parallel, and high-throughput manner will be critical to the optimization of microbial cell factories.

The clustered regularly interspaced short palindromic repeats (CRISPR) system has been recently developed for multiplex genome engineering in nearly all kingdoms of life[7–10]. In this modular and highly efficient system, a guide RNA (gRNA) recruits a CRISPR nuclease to a specific region of the genome to create in a double strand break. Moreover, transcriptional activation or interference can be achieved by fusing an activation or repression domain to the nuclease-deficient CRISPR protein[11–15]. Although the CRISPR system has been demonstrated with the capability of transcriptional activation (CRISPRa), transcriptional interference (CRISPRi), and gene deletion (CRISPRd), there have been few studies aiming to develop a multi-functional CRISPR system. One such effort took advantage of the orthogonal RNA scaffolds[15], where aptamers were added to the gRNA and the corresponding RNA binding protein recruited a functional effector domain, such as an activation domain for CRISPRa and a repression domain for CRISPRi. Another study[16, 17] was based on the finding that the truncated gRNA could still bind to target DNA sequences but without introducing a double strand break. In other words, the truncated gRNA can be used for transcriptional regulation and the full-length gRNA is used as a nuclease for genome editing, which enabled simultaneous gene activation, repression, and deletion using a single Cas9 protein. However,

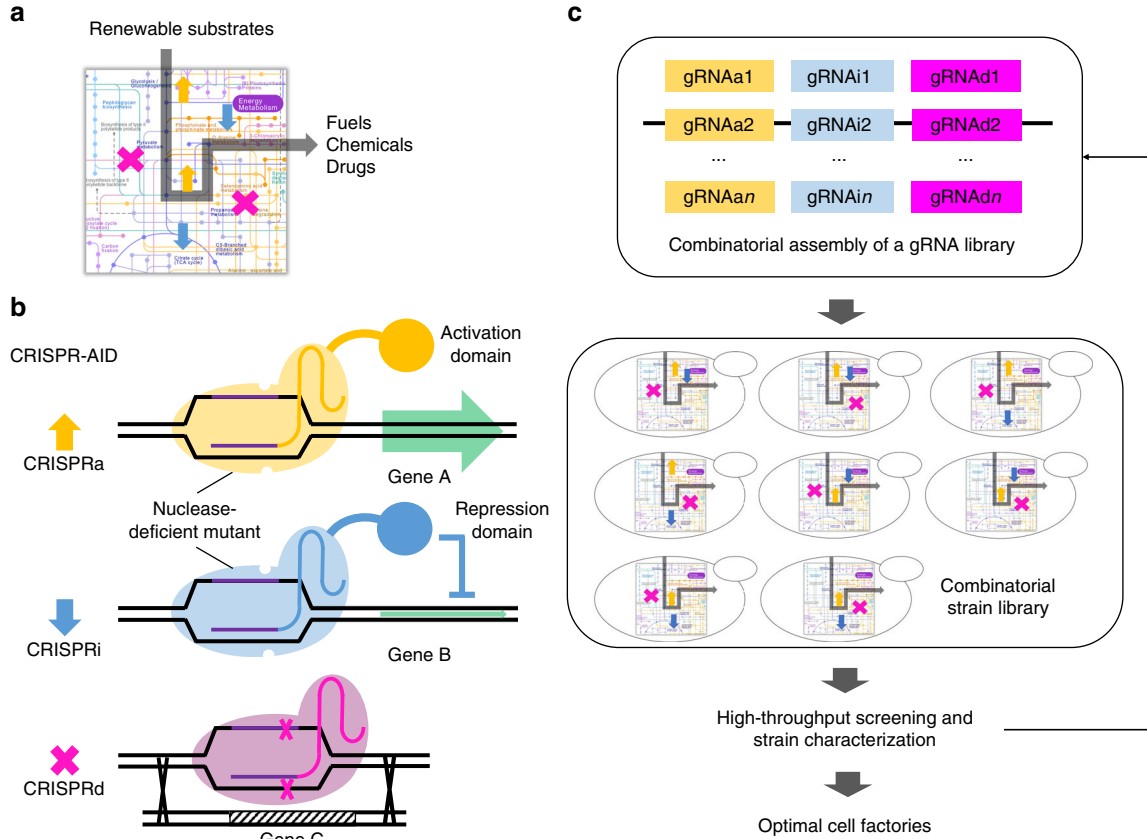

**Fig. 1** Design of CRISPR-AID for combinatorial metabolic engineering. **a** Cell factories for sustainable production of fuels, chemicals, and drugs from renewable resources. **b** Development of CRISPR-AID using three orthogonal CRISPR proteins, one nuclease-deficient CRISPR protein fused with an activation domain for CRISPRa, another nuclease-deficient mutant fused with a repression domain for CRISPRi, and a third catalytically active CRISPR protein for CRISPRd. **c** CRISP-AID enables combinatorial metabolic engineering by exploring all the possible gRNA combinations to construct optimal cell factories

**Table 1 Nuclease activity of CRISPR protein orthologs in yeast**

| Nuclease | gRNA | Protein-NLS | NLS-Protein-NLS |
|---|---|---|---|
| SpCas9 | Sg10 | – | ~ 80% |
| NmCas9 | Sg12 | – | 0 |
| | Sg13 | – | 0 |
| | Sg29 | – | 0 |
| | Sg32 | – | 0 |
| St1Cas9 | Sg14 | 0 | ~ 62% |
| | Sg15 | 0 | 0 |
| | Sg30 | 0 | ~ 2.4% |
| | Sg64 | 0 | ~ 72% |
| SaCas9 | Sg31 | ~ 50% | ~ 46% |
| | Sg93 | ~ 27% | ~ 30% |
| | Sg94 | ~ 4.6% | ~ 5.2% |
| | Sg95 | ~ 77% | ~ 84% |
| AsCpf1 | Sg68 | 0 | ~ 0.2% |
| LbCpf1 | Sg69 | 0 | ~ 59% |
| | Sg122 | 0 | ~ 92% |
| | Sg123 | 0 | ~ 55% |
| | Sg124 | 0 | ~ 0.3% |

Nuclease activity was evaluated by co-transforming 500 ng CRISPR protein plasmid, 500 ng gRNA plasmid, and 500 ng linear DNA donor for the deletion of whole *ADE2* coding sequences. The results represent an average of biological triplicates

such a multi-functional CRISPR system was only possible when purposely designed synthetic promoters (i.e., CRISPR-repressible promoters and CRISPR-activatable promoters) were used, indicating a lack of general applicability. In addition, the potential competition between different gRNAs for the same Cas9 protein may be another major concern for metabolic engineering applications. Therefore, we aim to develop an orthogonal and generally applicable tri-functional CRISPR system comprising CRISPRa, CRISPRi, and CRISPRd (CRISPR-AID) for metabolic engineering of *Saccharomyces cerevisiae*, one of the prominent microbial cell factories for industrial applications. Due to the modular and multiplex advantages of the CRISPR system, CRISPR-AID can be used for combinatorial optimization of various metabolic engineering targets and exploration of the synergistic interactions among transcriptional activation, transcriptional interference, and gene deletion in *S. cerevisiae*.

In the present study, we develop a CRISPR-AID system using three orthogonal CRISPR proteins, one for activation, one for interference, and one for deletion. Since only one CRISPR protein has been well characterized for genome engineering in yeast[15, 18–23], we first characterize a number of CRISPR protein orthologs. Those functional CRISPR proteins are further optimized for transcriptional regulation by engineering the optimal effector domains. Using the optimized CRISPR-AID system, 5-fold activation of a red fluorescent protein, 5-fold interference of a yellow fluorescent protein, and >95% deletion of an endogenous gene is achieved simultaneously by transforming a single plasmid into yeast. We then demonstrate the application of CRISPR-AID for rational metabolic engineering with β-carotene production as a case study. Finally, we apply CRISPR-AID for combinatorial optimization of the metabolic engineering targets to enhance the expression and display of a recombinant protein on the yeast surface by 2.5-fold as well as exploring the synergistic interactions among these genomic modifications.

## Results

**Design of CRISPR-AID for combinatorial metabolic engineering.** To construct optimal cell factories using combinatorial metabolic engineering, we need a synthetic biology toolkit that enables different modes of genetic manipulation of multiple targets in the metabolic and regulatory network, including increased expression, decreased expression, and zero expression, in a modular, parallel and high-throughput manner (Fig. 1a). In this study, we developed a tri-functional CRISPR-AID system using three orthogonal CRISPR proteins (Fig. 1b), one nuclease-deficient CRISPR protein fused with an activation domain for transcriptional activation (CRISPRa), a second nuclease-deficient CRISPR protein fused with a repression domain for transcriptional interference (CRISPRi), and a third catalytically active CRISPR protein for gene deletion (CRISPRd). For metabolic engineering of complex phenotypes, such as stress tolerance and production of recombinant proteins, we can identify numerous metabolic engineering targets. Since the host genome can be manipulated in a modular and high-throughput manner via plasmid-borne gRNAs, CRISPR-AID enables combinatorial optimization of various metabolic engineering targets. In conjugation with high-throughput screening, we will be able to obtain the combination of the activated, interfered, and deleted metabolic engineering targets that work synergistically to yield the optimal phenotype (Fig. 1c). If necessary, the process can be repeated iteratively.

**Construction and optimization of the CRISPR-AID system.** To enable fast evaluation of orthogonal genome editing and transcriptional regulation, we constructed a reporter yeast strain: *mCherry* driven by a medium-strength promoter *CYC1p* for CRISPRa, *mVenus* driven by a strong promoter *TEF1p* for CRISPRi, and *ADE2*, an endogenous gene whose disruption would result in the formation of red colonies in adenine deficient synthetic medium, for CRISPRd (Supplementary Fig. 1). Since only SpCas9 has been well characterized in yeast, we included dSpCas9-VPR[24], dSpCas9-MXI1[23], and SpCas9[19, 20] as the positive controls for the optimization of CRISPR-AID modules. When tested individually in the reporter yeast strain CT, we obtained more than 5-fold activation of *mCherry* expression (dSpCas9-VPR with Sg6), around 10-fold interference of *mVenus* expression (dSpCas9-MXI1 with Sg1), and nearly 100% deletion of *ADE2* gene (SpCas9 with Sg11) (Supplementary Fig. 1).

To develop the orthogonal tri-functional CRISPR system, at least three functional CRISPR proteins are needed. Thus, we sought to characterize a few CRISPR protein orthologs in *S. cerevisiae*. In previous studies, several CRISPR proteins (Supplementary Table 1) including Cas9 from *Streptococcus pyogenes* (SpCas9)[8, 9, 20], *Neisseria meningitides* (NmCas9)[25, 26], *Streptococcus thermophiles* (St1Cas9)[26, 27], and *Staphylococcus aureus* (SaCas9)[27, 28] and Cpf1[29] from *Lachnospiraceae bacterium* ND2006 (LbCpf1) and *Acidaminococcus sp.* BV3L6 (AsCpf1) have been characterized and found to be functional in mammalian cells. Therefore, we firstly characterized the nuclease activities of these CRISPR proteins in yeast, using *ADE2* deletion as a reporter. Interestingly, although a single nuclear localization sequence (NLS) tag at the C-terminus was sufficient to target the CRISPR proteins to the nucleus of mammalian cells[26, 27, 29], we found that dual-NLSs at both termini were required for nuclease activity of St1Cas9 and LbCpf1 in yeast (Table 1). We were unable to detect any nuclease activity for NmCas9 and AsCpf1 under any conditions, probably due to different protein folding environments between yeast and mammalian cells. We characterized more than three CRISPR proteins to be functional and orthogonal to each other, i.e. functional only when bound to their own cognate gRNAs (Supplementary Fig. 2). To enable multiplex genome engineering, we followed the previously developed HI-CRISPR design[20], where the homology donor sequences were integrated into the gRNA expression cassette. We found that the stable maintenance of the homology donor resulted in a further

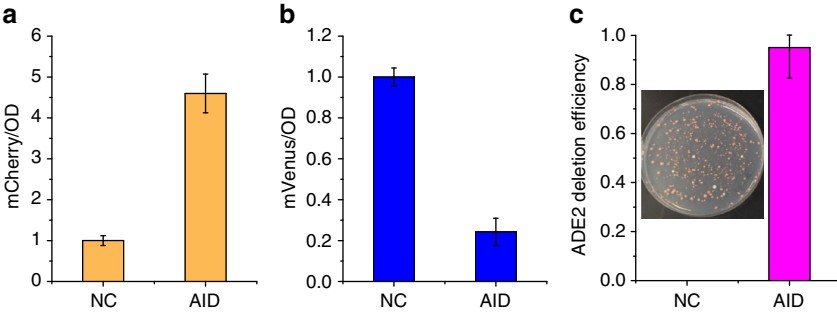

**Fig. 2** Demonstration of CRISPR-AID using the reporter yeast strain CT. By transforming the reporter strain with a single plasmid containing an array of 3 gRNAs, transcriptional activation of *mCherry* **a**, transcriptional interference of *mVenus* **b**, and deletion of an endogenous *ADE2* gene **c** were achieved simultaneously with high efficiency. The inset in **c** shows a representative result of *ADE2* deletion using CRISPR-AID. Error bars represent the mean ± s.d. of biological quadruplicates

increase in CRISPRd efficiency: from 80% with Sg10 (Table 1) to ~98% with Sg11 (Supplementary Fig. 1) for SpCas9 and from 77% (Table 1) with Sg95 to ~95% with Sg145 for SaCas9. Therefore, SpCas9, SaCas9, St1Cas9, and LbCpf1 as well as their corresponding nuclease-deficient forms were chosen for further studies.

We then optimized the combination of the CRISPR proteins and the activation domains to achieve maximal CRISPRa. By testing all possible combinations (Supplementary Fig. 3a) of four nuclease-deficient CRISPR proteins (dSpCas9, dSaCas9, dSt1Cas9, and dLbCpf1) and three activation domains (VP64 (V), VP64-p65AD (VP), and VP64-p65AD-Rta (VPR))[24], we found that the optimal activation domain was CRISPR protein dependent: for dSpCas9, stronger activation domain resulted in more efficient CRISPRa (Supplementary Fig. 3b); for dSt1Cas9, the order was completely reversed (Supplementary Fig. 3d); while for dLbCpf1, the medium strength activation domain (VP) worked the best (Supplementary Fig. 3e). Interestingly, although SaCas9 was functional for CRISPRd, we only observed marginal activation using dSaCas9 with various activation domains and several gRNAs targeting different regions of *CYC1p* and *RNR2p* (Supplementary Fig. 3c). Since only 1 out of 12 gRNAs resulted in significant transcriptional activation (Supplementary Fig. 3d), dSt1Cas9 was not further evaluated for practical metabolic engineering applications. Therefore, we chose dSpCas9 and dLbCpf1 as CRISPRa candidates.

In previous studies, only one repression domain from mammalian cells (MXI1) has been reported and used for CRISPRi in yeast[23]. We hypothesized that it might not be optimal and the endogenous repression domain should work better to achieve maximal CRISPRi (Supplementary Fig. 4a). CRISPRi can be achieved by either blocking transcriptional initiation (i.e., binding to the promoter region) or transcriptional elongation (i.e., binding to the coding sequences). Indeed, although dSpCas9-MXI1 could block transcriptional initiation efficiently, the CRISPRi efficiency to block transcriptional elongation was much lower (Supplementary Figs. 4 and 5). By replacing MXI1 with the native repression domains (Supplementary Table 2), such as those from TUP1, MIG1, and UME6, the efficiency of CRISPRi was significantly improved. Among several repression domains, RD2, RD5, and RD11 worked the best when fused at the C-terminus of dSpCas9 for CRISPRi (Supplementary Fig. 4b). Inspired by the design of strong activation domains for CRISPRa[24], we combined multiple repression domains together, either in the form of N- and C-terminal tagged or tandem repeat at the C-terminus, to engineer an optimal repression domain for CRISPRi (Supplementary Fig. 4a). It was found that the use of multiple repression domains further enhanced CRISPRi efficiency (Supplementary Fig. 4c). More importantly, the engineered

repression domain also improved CRISPRi efficiency when targeting other promoters, such as *FBA1p* and *HHF2p* (Supplementary Fig. 6). dSpCas9-RD1152 (dSpCas9-RD11-RD5-RD2) demonstrated the highest CRISPRi efficiency and was chosen for further studies. Since dLbCpf1 was not efficient enough for CRISPRi (Supplementary Fig. 5b), we finalized the optimal design of the tri-functional and orthogonal CRISPR-AID system to be dLbCpf1-VP for CRISPRa, dSpCas9-RD1152 for CRISPRi, and SaCas9 for CRISPRd.

After optimization of the individual modules, we assembled all three CRISPR modules together and integrated them into the yeast genome for stable maintenance. In addition, an endoribonuclease (Csy4) module was included for multiplex processing of gRNAs. In this case, several gRNAs can be transcribed in a single expression cassette, if the Csy4 recognition sites are introduced between neighboring gRNA sequences. First, we cloned an array of 3 gRNAs downstream of *SNR52p* (design I), a type III promoter commonly used for gRNA expression in yeast. Unfortunately, only the first two gRNAs were found to be functional (Supplementary Fig. 7), probably due to the limited capability of the type III promoter to transcribe long sequences. Then we tested the expression of multiple gRNAs as individual expression cassettes (design II) or using a type II promoter (*TEF1p*, design III). In both cases, all the three gRNAs were fully transcribed and the tri-functional CRISPR-AID was demonstrated in the reporter yeast strain CT (Supplementary Fig. 7). As shown in Fig. 2, after introducing a single plasmid containing an array of gRNAs (pSg163, design III), the expression of *mCherry* was increased by 5-fold, the expression of *mVenus* was decreased by 5-fold, and the deletion of *ADE2* was achieved with an efficiency higher than 95%. More importantly, we obtained comparable CRISPRa, CRISPRi, and CRISPRd efficiencies when the gRNAs were cloned individually or in the array format (Supplementary Fig. 7). Notably, CRISPRi was demonstrated by targeting *mVenus* coding sequences (blocking transcriptional elongation) rather than targeting *TEF1p* (blocking transcriptional initiation), since the expression of SaCas9 and the gRNA array were both driven by *TEF1p* in our CRISPR-AID system. Otherwise, much higher CRISPRi efficiency could be expected by slightly modifying the design of the gRNA array.

**Rational metabolic engineering using CRISPR-AID**. After the proof-of-concept study, we sought to confirm that CRISPR-AID can be stably maintained and used for metabolic engineering applications. To this end, we tested CRISPR-AID with a well-known phenotype, the production of β-carotene in yeast. In previous studies, it has been found that overexpression of *HMG1*[30, 31], encoding a rate-limiting enzyme of the mevalonate

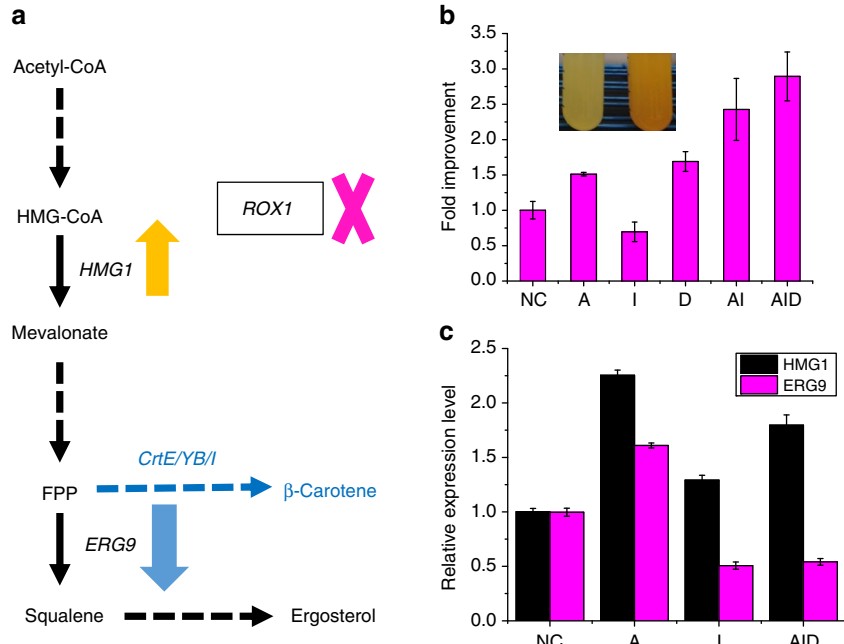

**Fig. 3** CRISPR-AID for rational metabolic engineering. **a** β-Carotene biosynthesis as a representative example of rational metabolic engineering. *HMG1*, *ERG9*, and *ROX1* were chosen as the targets for CRISPRa, CRISPRi, and CRISPRd, respectively. **b** Improved β-carotene production using single gRNA plasmids (A-pSg175, I-pSg172, and D-pSg186), a double gRNA plasmid (AI-pSg585), and a triple gRNA plasmid (AID-pSg239). The inset shows the yeast cultures before (SgH) and after (Sg239) CRISPR-AID engineering. **c** Verification of CRISPRa (*HMG1*) and CRISPRi (*ERG9*) for transcriptional regulation using qPCR. Error bars represent the mean ± s.d. of biological triplicates

pathway, down-regulation of *ERG9*[30], an essential gene at the branching point of the β-carotene biosynthesis and endogenous sterol biosynthesis, and the deletion of *ROX1*[32], encoding a stress responsive transcriptional regulator, could significantly increase the production of β-carotene. Therefore, we chose these three targets for CRISPRa, CRISPRi, and CRISPRd, respectively (Fig. 3a). Indeed, we found that single gRNA resulted in around 1.7-fold improvement in β-carotene production, while the combination of three gRNAs further improved the production to 2.8-fold (Fig. 3b). Quantitative PCR (qPCR) and diagnostic PCR further confirmed the enhanced expression of *HMG1*, down-regulation of *ERG9* (Fig. 3c), and deletion of *ROX1* (Supplementary Fig. 8a). Notably, the overexpression of *HMG1* resulted in increased expression of *ERG9*, probably due to the enhanced overall metabolic fluxes towards the mevalonate pathway. In addition, the repression of *ERG9* lowered the production of β-carotene, probably due to impaired cell fitness. In other words, *HMG1* up-regulation and *ERG9* down-regulation should be combined to achieve high β-carotene production (Fig. 3b). Such a synergy between up-regulation of *HMG1* and down-regulation of *ERG9* was consistent with previous studies[33].

**CRISPR-AID for combinatorial metabolic engineering.** Finally, we applied CRISPR-AID for combinatorial metabolic engineering. We chose recombinant protein expression via yeast surface display because the entire biological process is very important but rather complicated: proteins are translated in the cytosol, folded in the ER, glycosylated in the Golgi, and sorted and secreted to different compartments, and finally attached to the yeast cell surface (Fig. 4a). Many engineering targets have been explored[34], including the up-regulation of the secretory pathway and down-regulation of the protein degradation and competing pathways, although they have been mainly tested individually. Using CRISPR-AID, we can explore the gain-of-function and loss-of-

function combinations that work synergistically to increase recombinant protein displaying levels. Here, we chose *Trichoderma reesei* endoglucanase II (EGII) as the protein of interest[35], and 14 targets for CRISPRa, 17 targets for CRISPRi, and 5 targets for CRISPRd (Supplementary Table 3), most of which increased EGII display levels when tested individually (Supplementary Fig. 9). We then generated a library consisting of all the possible combinations (15×18×6 = 1620). Genotyping of several randomly picked colonies indicated that all plasmids were assembled correctly and the library was representative (Supplementary Table 4). Since the proteins are expressed on the yeast surface, we used an antibody conjugated with a fluorescent dye to detect the epitope tag and convert protein expression levels to fluorescence signals (Supplementary Figs. 10 and 11). Increased EGII activity of the sorted library using Fluorescence Activated Cell Sorting (FACS) indicated that the protein display levels were positively correlated with the fluorescence intensities (Supplementary Fig. 12). By enriching the highly fluorescent yeast cells, we obtained a few combinations that increased the protein expression levels and EGII activities significantly (Supplementary Fig. 13). Through DNA sequencing, we found that the interference and deletion targets were highly enriched, and the two clones showing the highest cellulase activity shared the same combination (Supplementary Table 5). Therefore, the combination of *PDI1* up-regulation, *MNN9* down-regulation, and *PMR1* deletion increased EGII display levels and cellulase activity the most (Fig. 4b). The increased expression of *PDI1* (CRISPRa) and decreased expression of *MNN9* (CRISPRi) were further confirmed using qPCR (Fig. 4c), and the deletion of *PMR1* (CRISPRd) at high efficiency was verified by diagnostic PCR (Supplementary Fig. 8b).

Interestingly, none of the components (*PDI1* activation, *MNN9* interference, and *PMR1* deletion) of the best combination increased EGII display level the most in each category when tested individually, indicating possible synergistic interactions among these genomic modifications. To figure out the potential

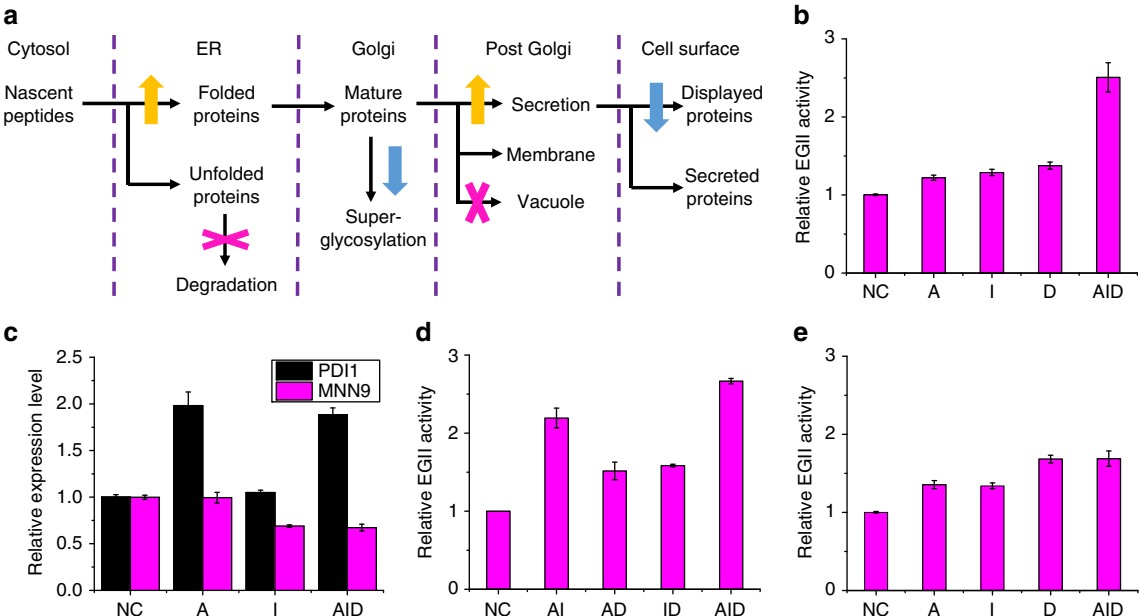

**Fig. 4** CRISPR-AID for combinatorial metabolic engineering. **a** Yeast surface display of recombinant proteins as a representative example of combinatorial metabolic engineering. Protein folding and secretory machinery, protein super-glycosylation and other surface-displayed proteins, and degradation pathways were chosen as the targets for CRISPRa, CRISPRi, and CRISPRd, respectively. **b** Combinatorial optimization of EGII display on the yeast surface. EGII activities of the FACS enriched optimal combination (AID-FACS16) and those with the corresponding single component (A-pSg221, I-pSg230, and D-pSg205) were measured. **c** Verification of CRISPRa (*PDI1*) and CRISPRi (*MNN9*) for transcriptional regulation using qPCR. **d** Exploration of the synergistic interactions among activated (*PDI1*), interfered (*MNN9*), and deleted (*PMR1*) metabolic engineering targets. EGII activities of the double mutants, including AI-pSg417, AD-pSg418, and ID-pSg419, were measured. **e** Single-factor optimization of EGII display on the yeast surface. EGII activities of the strains with one gRNA (A-pSg218, I-pSg204, and D-pSg186) and the combination of the ones with the highest activities in each category (AID-pSg257) were measured. Error bars represent the mean ± s.d. of biological triplicates

synergistic interactions, we constructed all the double mutants, including AI (*PDI1* activation and *MNN9* interference), AD (*PDI1* activation and *PMR1* deletion), and ID (*MNN9* interference and *PMR1* deletion), and measured their cellulase activities. As shown in Fig. 4d, we observed a clear synergistic interaction between *PDI1* activation and *MNN9* interference to increase the protein display levels and EGII activities, but not between the activation and interference targets and the deletion target. *PDI1* encodes a protein disulfide isomerase, which is essential for disulfide bond formation in secretory and cell-surface proteins. *MNN9* encodes a subunit of Golgi mannosyl-transferase complex, which mediates elongation of the poly-saccharide mannan backbone and involves in *N*-glycosylation of the native and recombinant proteins. A previous study[36] found that the deletion of *MNN9* increased the expression of a couple of genes related to protein secretion, but did not induce the unfolded protein response, such as the expression of *PDI1*, which might explain the synergy between *PDI1* overexpression and *MNN9* down-regulation for recombinant protein secretion and display. Finally, we compared combinatorial optimization with the traditionally used single-factor optimization for metabolic engineering applications, where the top candidates from each category (*ERO1* activation, *PMR1* interference, and *ROX1* deletion) were combined. As shown in Supplementary Fig. 14, transcriptional regulation of *ERO1* and *PMR1* and genome editing of *ROX1* were verified by qPCR and diagnostic PCR, respectively. Unfortunately, we observed no positive effects by combining these three metabolic engineering targets together (Fig. 4e), indicating the significance of combinatorial optimization of cellular metabolism and the advantage of CRISPR-AID to explore the synergy of various metabolic engineering targets for microbial cell factory development.

## Discussion

Microbial biosynthesis generally involves the introduction of a heterologous biosynthetic pathway to convert cellular metabolic precursor(s) to the desired product of interest[1]. To maximize product titers, yields, and productivities for commercially viable processes, the metabolic and regulatory networks of the producing microorganisms must be rewired (genome engineering)[3–5], i.e., to enhance the supply of the precursor metabolites, and the expression levels of the heterologous pathways must be carefully balanced (pathway engineering)[37], i.e., to avoid the accumulation of some toxic intermediates. While significant progress in multi-gene pathway optimization has been achieved, such as DNA copy number tuning[21, 38, 39], promoter engineering[37, 40], intergenic region engineering[41], terminator engineering[42], ribosome binding site engineering[12], and dynamic metabolic flux balancing[43], genome engineering to rewire the host cell metabolism is much more challenging. Therefore, the development of novel genome engineering tools to modify the host genome in a modular, parallel, and high-throughput manner will shorten the gap between pathway engineering and genome engineering and is essential for the optimization of microbial cell factories. Currently, MAGE (multiplex automated genome engineering) still represents the most successful technology for combinatorial optimization of multiple metabolic engineering targets, where short and synthetic oligonucleotides were used to simultaneously introduce genetic modifications at multiple loci of the *Escherichia coli* genome in an automated and iterative manner. More specifically, degenerate ribosome binding site (RBS) sequences flanked by homology arms were synthesized as oligonucleotide pools to fine-tune the expression level of the whole 1-deoxy-D-xylulose-5-phosphate (DXP) biosynthetic pathway to maximize lycopene production[44]. MAGE has also been combined with TRMR (tractable multiplex recombineering)[45, 46] for combinatorial genome engineering in *E.*

*coli*. Nevertheless, the application of MAGE in *S. cerevisiae*, a preferred host for industrial production of fuels, chemicals, and pharmaceutics, is less successful, probably due to the lower efficiency of short oligonucleotides mediated allelic replacement[47]. In addition, since the transcription and translation machinery (i.e., RBS sequences) is much more complicated and less understood in eukaryotes, short oligo-mediated genome modification may not be sufficient to fine-tune the expression level of the targeted genes for combinatorial metabolic engineering in yeast. To address these challenges, a new strategy was recently developed by combining cDNA overexpression and RNA interference (RNAi) for combinatorial genome-scale engineering of complex phenotypes (i.e., acetic acid tolerance and glycerol utilization) in yeast[35]. In this study, we developed CRISPR-AID, a tri-functional CRISPR system combining transcriptional activation, transcriptional interference, and gene deletion, for combinatorial metabolic engineering in yeast. Both strategies enable the exploration of the gain- and loss-of-function combinations that work synergistically to improve the desired phenotypes. Nevertheless, CRISPR-AID not only introduces a third mode of genome engineering (gene deletion), but also has different mechanisms of genome modulation and offers several advantages. For example, down-regulation using CRISPRi or RNAi is required for the modulation of essential genes, while CRISPRd enables more stable and in many cases significant phenotypes when targeting non-essential genes; CRISPRa is less biased for overexpression of large genes during large scale combinatorial optimization[11, 14]; CRISPRi blocks transcription in the nucleus while RNAi affects mRNA stability and translation, and CRISPRi is generally found to have higher repression efficiency in many situations[11, 14]. Using CRISPR-AID, we can introduce different modes of genomic modifications (i.e., activation, interference, and deletion) via gRNAs on a plasmid. Similar to combinatorial pathway optimization by exploring all the possible gene expression levels[37], we can achieve combinatorial metabolic engineering by testing all the possible gRNA combinations (Fig. 1c). In this case, we can explore all the combinations of the metabolic engineering targets of the metabolic and regulatory network related to the desired phenotype.

As mentioned above, although the CRISPR-based genome engineering technology has grown exponentially in recent years, most of the current studies mainly focus on a mono-function CRISPR in a specific biological system. The recently developed dual-functional CRISPR systems were based on RNA scaffolds (CRISPRa and CRISPRi)[15] and gRNA truncation (CRISPRa and CRISPRd)[16, 17]. Our initial design of a tri-functional CRISPR system was to combine these two strategies: truncated gRNA with the MS2 aptamer to recruit MS2-VP64 for transcriptional activation, truncated gRNA with the Com aptamer to recruit Com-MXI1 for transcriptional interference, and full-length gRNA for gene deletion. Compared with that of the full-length gRNA, we found that truncated gRNAs (12–16 nt targeting sequences) resulted in comparable CRISPRi (Supplementary Fig. 15) and CRISPRa (Supplementary Fig. 16) efficiency in yeast. In addition, the use of truncated gRNA together with modular RNA scaffold engineering (SpCas9 + Sg45 + MS2-VP64) worked equally well as one of the optimal CRISPRa designs (dSpCas9-VPR + Sg33 or Sg6). Unfortunately, CRISPRi efficiency was dramatically decreased when an aptamer was added to the gRNA scaffold, which might result from lower binding affinity between Cas9 and the engineered gRNA. The change of repression domains and the use of another aptamer-RNA binding domain pair did not significantly improve CRISPRi efficiency either (Supplementary Fig. 17). Interestingly, although orthogonal transcriptional regulation was developed using modular RNA scaffolds, the use of such a system for CRISPRi was only demonstrated in mammalian cells and un-modified gRNA (without aptamer and RNA binding protein to recruit a repression domain) was used for CRISPRi in yeast[15]. A most recent study following a similar design (gRNA with the MS2 aptamer to recruit MS2-VPR for transcriptional activation and gRNA with the PP7 aptamer to recruit PCP-MXI1 for transcriptional interference) resulted in limited success in transcriptional reprogramming and metabolic engineering applications in yeast[48]. In both cases, gRNAs were modified to be independent of each other to enable a dual-functional CRISPR system, while the Cas9 protein remained intact[15–17]. In other words, they are not fully orthogonal CRISPR systems, since competition between different gRNAs may still occur. Overall, a simple combination of the modular RNA scaffold engineering and the gRNA truncation strategies did not work to develop a tri-functional CRISPR system. In this study, we developed a fully orthogonal tri-functional CRISPR-AID by using three independent CRISPR proteins, whose protospacer adjacent motif (PAM) sequences and gRNA scaffold sequences are different from each other.

In the future, we will explore CRISPR-AID for genome-scale engineering, with potential applications in both metabolic engineering and fundamental studies. Although yeast is one of the most well studied microorganisms, we still do not have a clear and thorough understanding of the whole metabolic and regulatory networks. In previous metabolic engineering efforts, we often found that some unknown or unrelated targets resulted in the highest increase in the desired phenotype[49, 50]. Therefore, genome-scale metabolic engineering is needed to cover all the possible important targets. In the genome-scale CRISPR-AID system, we will create a comprehensive library that can control the expression of any single gene in the yeast genome to different levels (increased expression, decreased expression, and zero expression). Followed by high-throughput screening and next generation sequencing, multiple hits that increase the desired phenotype will be obtained, and the process can be repeated iteratively until the construction of optimal microbial cell factories.

In summary, we developed a tri-functional CRISPR-AID by combining transcriptional activation, transcriptional interference, and gene deletion in a single system, and applied CRISPR-AID for rational and combinatorial metabolic engineering. We also explored synergistic interactions among different genome modifications.

## Methods

**Strains and cultivation conditions**. *E. coli* strain DH5α was used to maintain and amplify plasmids and recombinant strains were cultured at 37 °C in Luria broth medium containing 100 μg mL$^{-1}$ ampicillin. *S. cerevisiae* CEN.PK2-1C strain (EUROSCARF, Frankfurt, Germany) was used as the host for homologous recombination based cloning, recombinant protein expression and surface display, and β-carotene production. Yeast strains were cultivated in complex medium consisting of 2% peptone and 1% yeast extract supplemented with 2% glucose (YPD). Recombinant strains were grown on synthetic complete medium consisting of 0.17% yeast nitrogen base, 0.5% ammonium sulfate, and the appropriate amino acid drop out mix, supplemented with 2% glucose (SCD). When necessary, 200 μg mL$^{-1}$ G418 (KSE Scientific, Durham, NC) was supplemented to the growth media. Ammonium sulfate was replaced with 0.1% mono-sodium glutamate (SED), when G418 was used in synthetic medium. All restriction enzymes, Q5 polymerase, and the *E. coli-S. cerevisiae* shuttle vectors were purchased from New England Biolabs (Ipswich, MA). All chemicals were purchased from Sigma-Aldrich (St Louis, MO) unless otherwise specified.

**Plasmid and strain construction**. Recombinant plasmids were constructed using restriction digestion/ligation, Gibson Assembly, Golden-Gate Assembly, or DNA assembler[51]. For the yeast homologous recombination based cloning, DNA fragments with homology arms at both ends were generated by PCR and co-transformed with the linearized vector into *S. cerevisiae*. Yeast plasmids were isolated using a Zymoprep Yeast Plasmid Miniprep II Kit (Zymo Research, Irvine, CA) and amplified in *E. coli* for verification by both restriction digestion and DNA sequencing. All the recombinant plasmids and gRNA plasmids used in this study

were listed in Supplementary Tables 6 and 7, respectively. Oligonucleotides used for gene amplification, pathway assembly, diagnostic PCR verification, and qPCR analysis were listed in Supplementary Table 8. Oligonucleotides and gBLOCKs (Integrated DNA Technologies, Coralville, IA, USA) used for gRNA construction were listed in Supplementary Tables 9 and 10, respectively. Yeast strains were transformed using the LiAc/SS carrier DNA/PEG method, and transformants were selected on the appropriate agar plates. Recombinant yeast strains constructed in this study were listed in Supplementary Table 11.

The reporter plasmid p406-CT was constructed by cloning each expression element including CYC1p, mCherry, TEF1t, TEF1p, mVenus, and PGK1t into pRS406 using Gibson Assembly. Other reporter plasmids were constructed by replacing TEF1p in p406-CT with FBA1p (strong promoter, p406-CF), HHF2p (strong promoter, p406-CH), REV1p (weak promoter, p406-CR1), and RNR2p (medium-strength promoter, p406-CR2), respectively. The reporter yeast strains were constructed by integrating EcoRV linearized reporter plasmids into the ura3 locus of the CEN.PK2-1C genome.

For the construction of individual gRNA expression plasmids, several helper plasmids (pSgH2, pSpSgH, pNmSgH, pSt1SgH, pSaSgH, pSpMS2SgH, pSpPP7SgH, and pSpComSgH) containing SNR52p, two BsaI sites, gRNA scaffold sequences, and SUP4t were constructed first based on a modified, BsaI-free pRS423 vector[20]. Then the targeting sequences were synthesized as short oligos, which were annealed and phosphorylated and cloned into the corresponding BsaI digested helper plasmids. To construct multiple gRNAs expression plasmids, the individual gRNA expression cassettes were pieced together using Golden-Gate Assembly (design II), or the gRNA arrays were synthesized as gBLOCKs and cloned into pRS423-H5 (design III) using restriction digestion/ligation.

CRISPR protein expression plasmids were constructed by cloning the PCR amplified fragments into pH1, pH3, pH4, pH5, and pH6[52, 53] using BamHI/XhoI or NcoI/XhoI digestion and ligation. To clone additional NLS into the N-terminus of some CRISPR proteins, adapter (BamHI-NLS-BamHI or NcoI-NLS-NcoI) was inserted into the BamHI or NcoI site. The nuclease-deficient LbCpf1 (E832A) was created by overlap extension PCR and cloned into the NcoI/HindIII site of pTDH3-dSpCas9-MXI1 to construct pTDH3-dLbCpf1-MXI1. MXI1 fragment of pTDH3-dSpCas9-MXI1 and pTDH3-dLbCpf1-MXI1 was replaced by HindIII/XhoI digestion to construct dSpCas9 with different repression domains and dLbCpf1 with various activation domains, respectively. pAID6 was constructed by cloning each CRISPR-AID module (dLbCpf1-VP, Csy4, dSpCas9-RD1152, and SaCas9) into pRS41K-CEN-Delta using DNA Assembler. CEN-iAID6 was constructed by integrating PmeI digested pAID6 into the delta site and selection for G418 resistance. The successful integration of AID6 cassettes was verified by both diagnostic PCR and CRISPR functional assays.

The β-carotene producing strain (CEN-Crt) and Trichoderma reesei endoglucanase II (EGII)-displaying strain (CEN-EGII) were constructed by integrating StuI linearized YIplac211-YB/E/I[31] and p406-YD-EGII (TEF1p-prepro-HisTag-EGII-AGA1-PGK1t)[35], respectively, into the ura3 locus of CEN-iAID6 genome and selection on SED-URA/G418.

**Fluorescence intensity measurement.** Recombinant yeast strains were pre-cultured in the corresponding selective medium for 2 days and then inoculated into the fresh synthetic media with an initial OD of 0.1. Mid-log phase yeast cells were diluted 5-fold in ddH$_2$O and mVenus and mCherry fluorescence signals were measured at 514–528 nm and 587–610 nm, respectively, using a Tecan Infinite M1000 PRO multimode reader (Tecan Trading AG, Switzerland). The fluorescence intensity (relative fluorescence units; RFU) was normalized to cell density that was determined by measuring the absorbance at 600 nm using the same microplate reader.

**β-Carotene production and quantification.** β-Carotene producing strains with gRNAs were pre-cultured in SED-HIS-URA/G418 medium for approximately 2 days, inoculated into 5 mL fresh medium with an initial OD$_{600}$ of 0.1 in 14 mL culture tubes, and cultured under aerobic conditions (30 °C, 250 rpm) for 5 days. The stationery phase yeast cells were collected by centrifuge at 13,000×g for 1 min and cell precipitates were resuspended in 1 mL of 3 N HCl, boiled for 5 min, and then cooled in an ice-bath for 5 min. The lysed cells were washed with ddH$_2$O and resuspended in 400 μL acetone to extract β-carotene. The cell debris was removed by centrifuge and the β-carotene containing supernatant was analyzed for its absorbance at 454 nm. The production of β-carotene was normalized to the cell density.

**Screening of EGII-displaying mutants.** After transforming the combinatorial gRNA library plasmids, the recombinant yeast strains (>10$^5$ independent clones with more than 100-fold redundancy) were cultured at 30 °C for 3 days and then subject to immunostaining and flow cytometry analysis[35]. The primary and secondary antibodies were monoclonal mouse anti-histidine tag antibody (1:100 dilution, Bio-Rad, Raleigh, NC, catalog # MCA1396GA) and goat anti-mouse IgG (H + L) secondary antibody, Biotin-XX conjugate (1:100 dilution, ThermoFisher Scientific, Rockford, IL, catalog # B-2763), respectively. The levels of biotin on the yeast surface were quantified using Streptavidin, R-phycoerythrin conjugate (1:100 dilution, ThermoFisher Scientific, catalog # S866). The phycoerythrin (PE)

fluorescence was analyzed with a LSR II Flow Cytometer (BD Biosciences, San Jose, CA). FACS experiments were performed on a BD FACS Aria III cell sorting system (BD Biosciences, San Jose, CA). In the first round of sorting, around 30,000 cells representing the top 1% highest fluorescence were collected. The second round of sorting collected 96 individual yeast cells with the highest fluorescence into a 96-well microplate. Then the plasmids were extracted and retransformed into the CEN-EGII strain with a fresh background, 26 of the retransformed yeast mutants conferred the highest PE fluorescence were further analyzed by the cellulase activity assay. Briefly, 400 μL yeast cells from overnight culture were washed twice with ddH$_2$O and resuspend in the same volume of 1% (w v$^{-1}$) carboxymethyl cellulose (CMC) solution (0.1 M sodium acetate, pH 5). After incubation at 30 °C for 16 h with vigorous shaking, the supernatant was analyzed using the modified DNS method[54] to quantify the amount of the reducing sugars, which was normalized to the cell density to represent the EGII enzyme activity.

**gRNA design.** gRNA for gene deletion (CRISPRd) was designed using Benchling CRISPR tool (https://benchling.com), and those with both high on-targeting score and off-targeting score were selected. For CRISPRa and CRISPRi, the gRNA binding position was equally important as the sequence itself. Based on previous studies[14, 55, 56] and our empirical experience, ~ 250 bp upstream of the coding sequences or ~ 200 bp upstream of the transcription starting site (TSS) worked the best for CRISPRa; ~ 100–150 bp upstream of the coding sequences or 50–100 bp upstream of TSS worked the best for CRISPRi by blocking transcriptional initiation and those targeting the non-template strand of the coding sequences worked the best for CRISPRi by blocking transcriptional elongation (Supplementary Fig. 5). Since on-targeting score and off-targeting score were not available for Cpf1, the following criteria were considered: GC contents between 35 and 65%, no polyT, no secondary structure, and minimal off-target effect (<12 bp match by BLAST to the yeast genome).

**Quantitative PCR analysis.** Mid-log phase yeast cells were collected and used to determine the relative expression levels via qPCR. Total RNAs were isolated using the RNeasy Mini Kit (QIAGEN, Valencia, CA, USA) following the manufacturer's instructions. In total 1 μg of the RNA samples were then reversed transcribed into cDNA using the Transcriptor First Strand cDNA Synthesis Kit using oligo-dT primer (Roche, Indianapolis, IN, USA). The qPCR experiments were carried out using SYBR Green-based method in the QuantStudio 7 Flex Real-Time PCR System (ThermoFisher Scientific).

**Data availability.** The authors declare that the main data supporting the findings of this study are available within the article and its Supplementary Information file or from the corresponding author upon reasonable request.

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

## Acknowledgements

This work was supported by the Carl R. Woese Institute for Genomic Biology at the University of Illinois at Urbana-Champaign and the U.S. Department of Energy (DE-SC0018260). J.L. also acknowledges the support of the Shen Postdoc Fellowship from the University of Illinois at Urbana-Champaign. We thank Dr Tong Si for his help and suggestions on yeast surface display and Dr Barbara Pilas from the Roy J. Carver Biotechnology Center for flow cytometry analysis and FACS.

## Author contributions

J.L. performed all the experiments and analyzed the data. J.L. and M.H. designed gRNAs. S.H. assisted with experimental work. J.L. and H.Z. conceived the study and wrote the manuscript.

## Additional information

**Competing interests:** The authors declare no competing financial interests.

