## [Peer Review File · Nature Communications]

Reviewers' comments:

Reviewer #1 (Remarks to the Author):

The manuscript reports the combinatorial metabolic engineering strategy for *Saccharomyces cerevisiae*, in which three orthogonal CRISPR proteins simultaneously and independently modulate transcriptional activation, transcriptional interference, and gene deletion.

Further, the authors demonstrate the usefulness of this strategy by applying to the improvement of carotenoid production and yeast surface display.

This strategy will be available as the tools for combinatorial metabolic engineering, however, the simultaneous and independently targeted gene regulation and editing with orthogonal Cas9 proteins or single Cas9 protein have been already reported [ref. #33 (Esvelt KM *et al*, Nat. Methods 10, 1116-1121 (2013)), #55 (Kiani, S. *et al*, Nat. Methods 12, 1051-1054 (2015)), and #56 (Dahlman JE *et al*, Nat. Biotechnol., 33, 1159-1161 (2015))]. Therefore, this reviewer believes that the system described in this manuscript does not advance the field significantly.

In general, the manuscript is well written and easy to understand.

A few exceptions that should be addressed are listed below.

Major point;

1. Page 4, lines 55-58, and 60-61

The statements about the novelty of the tri-functional CRISPR system may be a bit misleading at the moment, this needs to be re-formulated.

Perhaps, previous report [ref. 33 and 55] on other multifunctional CRISPR system using orthogonal Cas9 proteins could be introduced and discussed here.

2. Page 10, lines 202-206

The synergy effect between up-regulation of HMG1 and down-regulation of ERG9 is not clear to me.

The same β -caroten production experiment with "AI construct (sg175-sg172)" would be needed to clarify this point.

3. Page 12, lines 247-250

It is surprising that the authors found the synergistic interaction between two seemingly unrelated genes (PDI1 and MNN9).

However, this needs to be addressed as well. It would be useful to include the information about the function of PDI1, or a more detailed study would be needed to clarify this.

4. Page 12, lines 254-255

The authors observed no positive effects by combining the three targets (ERO1 activation, PMR1 interference, and ROX1 deletion) together.

It is even possible that there were little, if any, effect of CRISPR system on the expression levels of these three genes.

To clarify this, ERO1 activation, PMR1 interference, and ROX1 deletion should be confirmed by qPCR and diagnostic PCR.

Minor point

5. Figure 4d

What is the basis for the calculation of estimated EGII activity (Dashed line).

6. Supplementary Figure S2, legend

"gRNAs (Sg10, Sg64, Sg95, and Sg112)"

Perhaps Sg122, instead of Sg112.

7. Supplementary Table S7

pSg115-121 (related to Supplementary Figure S6) is missing.

Because lots of gRNAs were used throughout the paper, I would recommend the authors to add the schematic illustrations to summarize the design of gRNAs for each Figures (examples are found in LS Qi *et al*., Cell. 2013 Feb 28; 152(5): 1173–1183, Figure 7B).

Reviewer #3 (Remarks to the Author):

In the work titled: "Combinatorial Metabolic Engineering using an Orthogonal Tri-functional CRISPR System", Lian *et al* presents an approach based on a set of orthogonal CRISPR-Cas systems to elicit combinatorial genomic modifications in *S. cerevisiae*. The manuscript leverages a number of very recent and cutting edge advances in the characterization of novel Cas systems for mammalian genome editing for application in *S. cerevisiae* genome engineering. The authors describe the systematic characterization of a number of Cas systems for orthogonal CRISPR-mediated activation (CRISPRa), interference (CRISPRi), and deletion (CRISPRd), yielding a set of compatible systems. The authors then demonstrate the capacity of the system to generate combinatorial modification in activation, repression, and deletion in a number of gene targets for metabolic engineering of a recombinant protein which can be screened by yeast surface display. The results show that the optimal combinations of mutations could be quite unexpected based on single allele results, suggesting synergistic effects. This highlights that the combinatorial approach could be superior to that which can be derived from a more limited search of top single candidates using previous methodologies. Overall, the manuscript is clear, detailed, and concise. The experimental designs are sound and the results support the conclusions of the study. This work is timely and highlights the utility of combinatorial genome engineering using advanced CRISPR technologies that will be of particular value for the metabolic engineering community.

The reviewer has a few minor critiques:

1) The authors should perhaps better discuss/motivate scenarios in which CRISPRi vs CRISPRd is used and the advantages/disadvantages of using one over the other, or in combination. CRISPRi and CRISPRd are quite similar in the end phenotype result, especially in the context of screening for genetic modification to improve metabolic engineering goals. How is having both options useful, especially together?

2) The authors nicely demonstrated linked gRNA strategy (Design 3) using *cys4*-gRNAs architecture. What is the limit of the number of gRNAs that can be chained together? Does targeting the same loci with multiple gRNAs improve activation/repression levels? The gRNAa/gRNAi/gRNAd ratios are stoichiometrically coupled in Design 3. Can the ratios be tuned? Combinatorial tuning of not just binary activation/repression/deletion, but also the degree of each level would be very important in certain applications. Would this require the use of Design 2 using separate promoters of different activity levels for each gRNA set? What about the use of inducible promoters?

3) How does the optimization level of EGII via yeast surface display using CRISPR-AID compare with the best levels established in the literature using more traditional approaches?

4) Discussion of off-targeting effects and gRNA choices for efficient targeting in the system would improve the utility of CRISPR-AID for others in the field.

We thank all the reviewers for their thoughtful comments. We have fully addressed their questions and concerns point-by-point as described below. We have also followed the journal format requirement and editorial policy. All the corresponding changes are highlighted in yellow in the revised manuscript.

Reviewers' comments:

Reviewer #1 (Remarks to the Author):

The manuscript reports the combinatorial metabolic engineering strategy for *Saccharomyces cerevisiae*, in which three orthogonal CRISPR proteins simultaneously and independently modulate transcriptional activation, transcriptional interference, and gene deletion. Further, the authors demonstrate the usefulness of this strategy by applying to the improvement of carotenoid production and yeast surface display. This strategy will be available as the tools for combinatorial metabolic engineering, however, the simultaneous and independently targeted gene regulation and editing with orthogonal Cas9 proteins or single Cas9 protein have been already reported [ref. #33 (Esvelt KM et al, Nat. Methods 10, 1116-1121 (2013)), #55 (Kiani, S. et al., Nat. Methods 12, 1051-1054 (2015)), and #56 (Dahlman JE et al, Nat. Biotechnol.,33, 1159–1161 (2015))]. Therefore, this reviewer believes that the system described in this manuscript does not advance the field significantly. In general, the manuscript is well written and easy to understand. A few exceptions that should be addressed are listed below.

We appreciate the reviewer's comments and considering CRISPR-AID as a useful tool for combinatorial metabolic engineering. Nevertheless, we would like to argue the novelty of our current work.

1. Although dual functional CRISPR systems have been reported, there has been no attempt to develop a tri-functional CRISPR system, all of which are required for metabolic engineering applications. As described in this manuscript, a simple combination of previous strategies is not sufficient to develop an efficient tri-functional CRISPR system. The CRISPR proteins and effector domains must be carefully designed and engineered to achieve the optimal performance.
2. In addition, the previously reported dual-functional CRISPR systems were only demonstrated using reporters (such as fluorescent proteins), we applied CRISPR-AID for combinatorial metabolic engineering. As demonstrated in this manuscript, combinatorial optimization of the metabolic engineering targets is essential in developing optimal cell factories such as protein secretion and display on yeast surface.

Overall, we think we have developed a novel synthetic biology tool (CRISPR-AID) for novel applications (combinatorial metabolic engineering) based on the CRISPR system.

Major point;

1. Page 4, lines 55-58, and 60-61

The statements about the novelty of the ti-functional CRISPR system may be a bit misleading at the moment, this needs to be re-formulated. Perhaps, previous report [ref. 33 and 55] on other multifunctional CRISPR system using orthogonal Cas9 proteins could be introduced and discussed here.

We agree with the reviewer's comments. The introduction and discussion of other multifunctional CRISPR systems was provided in the Discussion section of the original manuscript and is moved to the Introduction section of the revised version (Line 56-65)

2. Page 10, lines 202-206

The synergy effect between up-regulation of HMG1 and down-regulation of ERG9 is not clear to me. The same β -caroten production experiment with “AI construct (sg175-sg172)” would be needed to clarify this point.

We appreciate the reviewer’s comments. To clarify the synergy between up-regulation of *HMG1* and down-regulation of *ERG9*, we tested β -carotene production using AI construct (Sg175-Sg172-SgH, **Fig. 3b**) and provided more discussion on the interaction of *HMG1* up-regulation and *ERG9* down-regulation (Line 207-213). In short, *HMG1* up-regulation and *ERG9* down-regulation should be combined to achieve high β -carotene production.

3. Page 12, lines 247-250

It is surprising that the authors found the synergistic interaction between two seemingly unrelated genes (*PDI1* and *MNN9*). However, this needs to be addressed as well. It would be useful to include the information about the function of *PDI1*, or a more detailed study would be needed to clarify this.

Thanks for the reviewer’s comments. The function of *PDI1* and *MNN9* is included in the revised manuscript (Line 252-256). Although *PDI1* and *MNN9* seem to be unrelated, they both involve in protein secretion process, *PDI1* for protein disulfide bond formation and *MNN9* for protein glycosylation. BTW, the endoglucanase used in the present study was reported to have disulfide bonds and be glycosylated. In addition, we provided a possible explanation for the synergistic interaction between *PDI1* up-regulation and *MNN9* down-regulation based on a previous study (Line 256-260).

4. Page 12, lines 254-255

The authors observed no positive effects by combining the three targets (*ERO1* activation, *PMR1* interference, and *ROX1* deletion) together. It is even possible that there were little, if any, effect of CRISPR system on the expression levels of these three genes. To clarify this, *ERO1* activation, *PMR1* interference, and *ROX1* deletion should be confirmed by qPCR and diagnostic PCR.

We appreciate the reviewer’s comments. To address the reviewer’s concern, *ERO1* activation, *PMR1* interference, and *ROX1* deletion are verified by qPCR and diagnostic PCR, respectively. As shown in Supplementary Fig. 14, the effects of the CRISPR system on the expression levels of these three genes are comparable among single gRNA plasmids (A-Sg218, I-Sg204, D-Sg186) and tri-gRNA plasmid (AID-Sg218-Sg204-Sg186) (Line 262-265 of the revised manuscript). These results confirm that the CRISPR system is still working and there is no synergy between the top candidates when tested individually, indicating the significance of combinatorial optimization of cellular metabolism using CRISPR-AID.

Minor point

5. Figure 4d

What is the basis for the calculation of estimated EGII activity (Dashed line).

Thanks for the review’s comments. The three dashed lines represent the estimated additive (no synergy) EGII activities with single (i.e. A), double (i.e. A+I), and triple modifications (i.e. A+I+D), respectively. Since the estimation is rather rough and not accurate, we decided to remove these dashed lined in Fig. 4d of the revised manuscript.

6. Supplementary Figure S2, legend

“gRNAs (Sg10, Sg64, Sg95, and Sg112)” Perhaps Sg122, instead of Sg112.

Thanks for the reviewer to point out the mistake. It has been corrected to Sg122 in the revised manuscript.

7. Supplementary Table S7

pSg115-121 (related to Supplementary Figure S6) is missing. Because lots of gRNAs were used throughout the paper, I would recommend the authors to add the schematic illustrations to summarize the design of gRNAs for each Figures (examples are found in LS Qi et al., Cell. 2013 Feb 28; 152(5): 1173–1183, Figure 7B).

We appreciate the reviewer's comments and suggestions. The plasmid info and primer sequences for Sg115-121 have been provided in the revised manuscript (Supplementary Table 7 and 9). We agree with the reviewer that schematic illustrations to summarize the design of gRNAs will be very helpful. Therefore, such gRNA design schemes are provided when multiple gRNAs target the same gene, particularly for CRISPRi and CRISPRa, where the position of targeting is equally important as the targeting sequence itself (Supplementary Figure S3, S4, S5, and S6).

Reviewer #3 (Remarks to the Author):

In the work titled: “Combinatorial Metabolic Engineering using an Orthogonal Tri-functional CRISPR System”, Lian et al presents an approach based on a set of orthogonal CRISPR-Cas systems to elicit combinatorial genomic modifications in *S. cerevisiae*. The manuscript leverages a number of very recent and cutting edge advances in the characterization of novel Cas systems for mammalian genome editing for application in *S. cerevisiae* genome engineering. The authors describe the systematic characterization of a number of Cas systems for orthogonal CRISPR-mediated activation (CRISPRa), interference (CRISPRi), and deletion (CRISPRd), yielding a set of compatible systems. The authors then demonstrate the capacity of the system to generate combinatorial modification in activation, repression, and deletion in a number of gene targets for metabolic engineering of a recombinant protein which can be screened by yeast surface display. The results show that the optimal combinations of mutations could be quite unexpected based on single allele results, suggesting synergistic effects. This highlights that the combinatorial approach could be superior to that which can be derived from a more limited search of top single candidates using previous methodologies. Overall, the manuscript is clear, detailed, and concise. The experimental designs are sound and the results support the conclusions of the study. This work is timely and highlights the utility of combinatorial genome engineering using advanced CRISPR technologies that will be of particular value for the metabolic engineering community.

Thanks very much for the reviewer's appreciation of our work. We believe CRISPR-AID will be a valuable synthetic biology tool for metabolic engineering.

The reviewer has a few minor critiques:

1) The authors should perhaps better discuss/motivate scenarios in which CRISPRi vs CRISPRd is used and the advantages/disadvantages of using one over the other, or in combination. CRISPRi and CRISPRd are quite similar in the end phenotype result, especially in the context of screening for genetic modification to improve metabolic engineering goals. How is having both options useful, especially together?

We appreciate the reviewer's comments. We briefly discussed the advantages and disadvantages of CRISPRi and CRISPRd in the revised manuscript (Line 307-310). Generally, CRISPRi is needed for engineering of essential genes, while CRISPRd is preferred to target non-essential genes, which gives clearer background and more stable phenotype. In most cases of metabolic engineering, we should modify both essential and non-essential genes, indicating the necessity of combining CRISPRi and CRISPRd.

2) The authors nicely demonstrated linked gRNA strategy (Design 3) using *cys4*-gRNAs architecture. What is the limit of the number of gRNAs that can be chained together? Does targeting the same loci with multiple gRNAs improve activation/repression levels? The gRNAa/gRNAi/gRNAd ratios are

stoichiometrically coupled in Design 3. Can the ratios be tuned? Combinatorial tuning of not just binary activation/repression/deletion, but also the degree of each level would be very important in certain applications. Would this require the use of Design 2 using separate promoters of different activity levels for each gRNA set? What about the use of inducible promoters?

Thanks for the reviewer's appreciation on our multiplex gRNA design.

1. The major goal of the present study is to show that CRISPR-AID is working and can be adopted for metabolic engineering applications. Therefore, we have not tried to express more than three gRNA in one cassette. Theoretically, type II promoter can generate large transcripts (> 10 kb), the limiting factor may be the activity of Csy4 and the stability of the long and complex transcript. It is an interesting and meaningful question, and we will try to figure it out in our future work.
2. Although not tested in the present study, previous studies have reported the improved activation/repression efficiency when targeting multiple gRNAs to the same gene, if the gRNAs are carefully designed (for example, the distance between each gRNA should be long enough to allow simultaneous and independent binding of the CRISPR protein).
3. We agree with the reviewer that fine-tuning of the CRISPR effects (activation/repression/deletion) is rather important in some metabolic engineering applications. Although not tested in the current manuscript, it can be achieved via
 - a) using gRNAs with different targeting efficiencies [Design II and III],
 - b) including different numbers of gRNAs for each target in the gRNA array [Design III],
 - c) using promoters with different strength [Design II], as suggested by the reviewer. However, type III promoters are not as well characterized as type II promoter. Thus, we plan to characterize more type III promoters for CRISPR-based metabolic engineering applications. Overall, we will further optimize CRISPR-AID system particularly for metabolic engineering applications in near future.
4. Actually, we have tested the use of inducible promoters for controllable expression of gRNA arrays, such as GAL1p (induced by galactose) and CUP1p (copper inducible). Unfortunately, the results are pretty much beyond our expectations, and we are trouble-shooting and doing more characterization now. We hope to get some interesting data for publication in near future.

3) How does the optimization level of EGII via yeast surface display using CRISPR-AID compare with the best levels established in the literature using more traditional approaches?

We appreciate the reviewer's comments. We did not compare EGII expression level achieved in this study with previous studies because the protein expression system including expression host, gene copy numbers, promoters, signal peptides, as well as fermentation conditions can be very different. To compete with the best levels established in the literature, we may have to perform iterative rounds of CRISPR-AID engineering, in conjunction with the optimization of gene copy numbers, signal peptides, and fermentation conditions. Since the major goal of the present study is to develop CRISPR-AID as a novel synthetic biology tool for combinatorial metabolic engineering, other metabolic engineering aspects are not pursued.

4) Discussion of off-targeting effects and gRNA choices for efficient targeting in the system would improve the utility of CRISPR-AID for others in the field.

We appreciate the reviewer's comments and suggestions. A paragraph focusing on gRNA design is provided in the Materials and Methods section in the revised manuscript (Line 468-479). Since the yeast genome is relatively small and the online gRNA design program has taken the off-targeting effect into consideration, we think the off-targeting effect should not be a major issue for metabolic engineering of yeast. Notably, we found that different gRNAs targeting the same gene tend to demonstrate similar phenotypes, confirming a minimal off-target effect in yeast when highly ranked gRNAs are used.

Finally, we would like to thank you and the reviewers again for thoughtful suggestions and comments, and we hope that the revised version of our manuscript meet the high standard of *Nature Communications* and will be accepted for publication. We are returning to you the revised manuscript.

Thank you very much for your interest and assistance.

Sincerely,

Huimin Zhao

REVIEWERS' COMMENTS:

Reviewer #1 (Remarks to the Author):

Basically, the authors replied well to all my comments and suggestions but there is one exception mentioned below.

- page 4, lines 65-67

This statements about the novelty of the tri-functional CRISPR system is an overstatement. As shown in the previous report (Figure 2, Kiani, S. et al., Nat. Methods 12, 1051-1054 (2015)), tri-functional CRISPR system for genome editing, activation and repression has been already reported, although the system was only demonstrated using reporters (such as fluorescent proteins).

Thus, I suggest the authors changing the sentence to highlight their first application of the tri-functional CRISPR system on metabolic engineering project.

Reviewer #3 (Remarks to the Author):

The authors have adequately addressed all of this reviewer's previous comments. The reviewer believes that the current manuscript represents a novel and important contribution to the field and is suitable for publication in Nature Communications.

-Harris Wang

Response to reviewers' comments:

Reviewer #1 (Remarks to the Author):

Basically, the authors replied well to all my comments and suggestions but there is one exception mentioned below.

- page 4, lines 65-67

This statements about the novelty of the tri-functional CRISPR system is an overstatement.

As shown in the previous report (Figure 2, Kiani, S. et al., Nat. Methods 12, 1051-1054 (2015)), tri-functional CRISPR system for genome editing, activation and repression has been already reported, although the system was only demonstrated using reporters (such as fluorescent proteins). Thus, I suggest the authors changing the sentence to highlight their first application of the tri-functional CRISPR system on metabolic engineering project.

We appreciate the reviewer's comments. As suggested by the reviewer, we briefly introduce the previous report on the development of a tri-functional CRISPR system using a single Cas9 protein and highlight that our tri-functional CRISPR system is orthogonal and generally applicable for metabolic engineering purposes. The corresponding changes are made on Page 4 Line 65-71 of the revised manuscript.

Reviewer #3 (Remarks to the Author):

The authors have adequately addressed all of this reviewer's previous comments. The reviewer believes that the current manuscript represents a novel and important contribution to the field and is suitable for publication in Nature Communications.

We appreciate this reviewer's help to improve our manuscript.